# ON AUGMENTING THE REFERENCES SECTION WITH A CITATION NETWORK VISUALIZATION

**Putra Manggala**
University of Amsterdam
p.manggala@uva.nl

**Tigran Atoyan**[*]
Dystematic Labs
tigran@dystematic.com

**Gracia Samosir**[*]
Fry eAssessment
grace@fry-it.com

**Jan Varsava**
Chartfleau
jan@chartfleau.com

**Johannes Ruf**
London School of Economics and Political Science
j.ruf@lse.ac.uk

## ABSTRACT

Researchers often find relevant articles by looking at the references section. We conducted user interviews with researchers about their workflow and their needs when carrying out literature research. Based on this study, we identify a set of problems encountered by researchers. We then propose to embed classified citation networks into articles as a solution, and to complement this graph with optional comments about references by authors. We demonstrate this idea by implementing it for this article. We argue that our solution helps increase inclusivity and improves the efficiency of reading scientific articles.

## 1 INTRODUCTION

In the field of Machine Learning (ML), the volume of conference submissions and the acceleration of scientific article publication as a means of dissemination of discovery are at an all time high. In order to uncover contemporary problems faced by researchers when reading scientific articles, we conducted user experience (UX) interviews with 20 researchers at various career stages (Section 2). From this study, we distilled a few key problems that were faced by our interviewees (Section 3). The proposed solution (Section 4) extends core aspects of research on citations over the past few decades, which we summarize below.

### 1.1 RELATED WORK ON CITATION TYPES AND NETWORK

**Citation types**  Various guidelines for reading articles (Purugganan & Hewitt, 2004; Keshav, 2007; Eisner, 2009) describe the importance of citations and of discovering related antecedent articles. Note that in this article we use the term *citation types* to capture different sets of semantics/classification labels for citations (i.e., a directed edge from a article to an antecedent article). Moravcsik & Murugesan (1975) first proposed four citation types according to their motivation (Table 1). Subsequently, Spiegel-Rosing (1977) expanded this set to thirteen different citation types. Teufel et al. (2006) and Han et al. (2016) used a subset of these citation types (see Table 2 for the former) and as labels for a citation classification task. Similarly, inspired by the "organic or perfunctory" citation type in Moravcsik & Murugesan (1975), Dong & Schäfer (2011) proposed four citation types which are deemed to be the most general and mutually exclusive and used them as labels (Table 3). There has also been a rich body of research on automated citation types classification (Xu et al., 2013; Bakhti et al., 2018), which helps assigning citation types at scale and standardizing the citation types to be assigned to citations.

**Citation network**  Citation types are commonly thought of as edge labels in a citation network. Various visualization tools have been built to aid with navigation in these networks. Some examples are CiteWiz (Elmqvist & Tsigas, 2007), Apolo (Chau et al., 2011), CircleView (Bergström & Whitehead Jr, 2006), and the tools proposed in Schäfer & Kasterka (2010) and Weitz & Schäfer

---

[*]Equal contribution. Sorted by last name.

(2012). In the user interviews we conducted, none of the researchers interviewed mentioned any of these tools to be part of their workflow. Chau et al. (2011) found a strong preference in using a network visualization component over a list format (i.e., a table) to display citation network data. Our proposed solution builds upon the concept of using spatial arrangement to manage the exploration of content when navigating citations.

| Binary citation types |
|---|
| Conceptual or operational use: use of theory vs. use of method. |
| Evolutionary or juxtapositional: work in the prime article is based on the cited work vs. an alternative to it. |
| Organic or perfunctory: work is critical for understanding and reproducing the citing article vs. a general acknowledgement out of politeness, policy, or piety as coined by Ziman (1968). |
| Confirmative or negational: the citing article confirms the cited article vs. disputes the cited article. |

Table 1: The four binary citation types discussed in Moravcsik & Murugesan (1975).

| Citation types | Description |
|---|---|
| **Weak** | Weakness of cited approach. |
| **CoCoGM** | Contrast/comparison in results (neutral). |
| **CoCo-** | Unfavourable contrast/comparison (current work is better than cited work). |
| **CoCoXY** | Contrast between two cited methods. |
| **PBas** | Author uses cited work as a starting point. |
| **PUse** | Author uses tools/algorithms/data. |
| **PModi** | Author adapts or modifies tools/algorithms/data. |
| **PMot** | This citation is positive about approach or problem addressed (used to motivate work in current article). |
| **PSim** | Author's work and cited work are similar. |
| **PSup** | Author's work and cited work are compatible or provide support for each other. |
| **Neut** | Neutral description of cited work, or not enough textual evidence for above categories or unlisted citation function. |

Table 2: The eleven citation types in Teufel et al. (2006) based on Spiegel-Rosing (1977).

## 1.2 REFLEXIVITY STATEMENT

We are a group of industry professionals from different sectors (finance, edTech, and e-commerce) and academics. Our motivation for conducting this study comes from the realization that navigating

| Citation types | Description |
|---|---|
| **Background** | Citations that describe background of the main topic on the whole, or provide recent studies and state-of-the-art approaches in a general way. |
| **Fundamental idea** | Citations about main previous work that inspired or gave specific hints on the current work. |
| **Technical basis** | Citations of important tools, methods, data, and other resources used or adapted in the current work. |
| **Comparison** | Citations comparing methods or results with the current work. |

Table 3: The four citation types proposed by Dong & Schäfer (2011).

and accessing the most relevant academic research output when working in the industry is challenging. To this end, we started interviewing researchers in academia and industry in order to understand their workflow in the hopes that we are able to learn how they are navigating and accessing research content. The citation network visualization is motivated by problems we discovered in our UX interviews with researchers.

## 2    USER EXPERIENCE INTERVIEWS

Over the course of three months, we conducted extensive contextual interviews with 20 researchers at various career stages. We focused on understanding the researchers' problems when they are conducting research.

### 2.1    RESEARCHER PROFILE SUMMARY

All 20 researchers were from fields with a computational component. Furthermore, at least one of the following requirements needed to be satisfied:

- The research area is interdisciplinary (e.g., Computational Neuroscience) or has applications within other disciplines (e.g., ML).
- The research area has industrial applications.
- The researcher has had past experiences in disseminating research knowledge to a non-expert audience.

The researchers interviewed came from diverse backgrounds:

- 10 interviewees had industry research experience.
- 6 interviewees were professors with experience in leading research groups and/or supervising graduate students.
- 10 interviewees were specialized in Computer Science/ML. The others came from various fields with a computational component and an interface with data science: Applied Mathematics, Computational Neuroscience, Computational Biology, Computational Physics, Biomedical Sciences, and Quantum Computing.

We tagged researchers according to the following roles (where a researcher can have multiple roles):

- Academics going into a new field (new researcher, **NR**).
- Industry researchers following academic research (industry researcher, **IR**).
- Research producers in academia (research producer, **RP**).

### 2.2    INTERVIEW FORMAT

Most interviews took an hour, with some taking less time due to time constraints of the interviewees, and some taking longer in the form of multiple calls with the same interviewee. The interviews

took place over a video call and were conducted by two interviewers: the interviewer with a better understanding of the interviewee's area would lead the interview, whereas the second interviewer would take notes used for post-call summary and analysis.

The interviewees were informed beforehand that the topic of the interview would be based on the broad question: *How can we make research more accessible with respect to navigating and understanding a research area?*[1]. The broad question served as a starting point for interviewees to express their thoughts and past experiences. We followed standard UX research guidelines of keeping questions open-ended and letting interviewees guide the discussion towards the problems they felt were most relevant[2].

Whenever a problem was brought up by researchers we interviewed, we tried to get to the root causes that triggered it. For example, "*I would like to read several articles on the same result.*" would be followed up by prompting what exactly the researcher is looking to establish by going through several versions of the result. We also directly asked the researchers to qualify the degree of inconvenience associated with a specified problem. This allowed us to identify the most relevant problems to address.

We also explored the current solutions / coping mechanisms the researchers might have. This was helpful in (1) identifying the incumbent best solutions, and (2) understanding whether the problem they expressed warrants a new solution. Interestingly, some of the problems stated were addressable by current tools, but they likely were not problematic enough to lead to the user's exploration of these existing tools.

## 3 INTERVIEW RESULTS AND KEY PROBLEMS

Based on the interview notes, we performed a card sorting exercise.[3] We noted the most common questions and issues stated by interviewees, and then categorized them into a high-level problem statement.

Throughout the interviews, these questions and issues (re-stated as questions) were often asked:

1. **(Q1)** What are the key articles to read for a review of the field?
2. **(Q2)** What articles are the most accessible and self-contained, in terms of giving an intuitive understanding without an overload of jargon and cross-references?
3. **(Q3)** When reading an article, what are the key reference articles to better understand the latter?
4. **(Q4)** What are the most relevant articles citing an article of interest?
5. **(Q5)** What are competing methodologies solving the same research problem addressed by a given research article?

We then categorized the questions and issues to the following high-level problem statements[4]:

- Not knowing what to read when exploring a new area of research (NR, IR).
- Inefficient consumption methods (reading texts/watching videos) of research works (NR, IR).
- Cumbersome ways to curate/track/archive explored literature (NR, IR, RP).

Researchers used various solutions to attempt to answer Q1-Q5 and therefore address the above problems, but none of the researchers interviewed mentioned being completely satisfied with them[5],

---

[1] This notion of "accessibility" is different from the notion we discuss in Section 6.

[2] See https://www.usability.gov/how-to-and-tools/methods/contextual-interview.html (retrieved on March 1, 2021). This website was developed by the U.S. Department of Health and Human Services to provide user experience best practices and guidelines.

[3] See https://www.usability.gov/how-to-and-tools/methods/card-sorting.html (retrieved on March 1, 2021).

[4] There are other questions/issues/high-level problem statements, but we chose to focus on a subset that we could target with the proposed solution.

[5] Solutions discussed during the interviews: ResearchGate, Google Scholar, arXiv, and Semantic Scholar.

and most found the tools to be of limited usability. We observed (1) "Tool Fatigue", where current solutions are deemed to be cumbersome to use as they are often third-party tools and are away from the reading experience, and (2) that current solutions cannot adequately help the interviewees answer Q1-Q5.

## 4 PROPOSAL TO HELP ANSWER Q1-Q5

### 4.1 EMBEDDING A CITATION NETWORK VISUALIZATION

In this position article, we hope to extend contemporary research in citation types and network in order to help answer Q1-Q5. Citation network visualization, along with citation types as edge labels, can be used to quickly and more accurately find out why an article is being cited. We would also like to propose the idea that the article itself should either store or point to the network visualization, which helps with the "Tool Fatigue" problem we observed. As an example, Figure 1 displays the citation network of Article 1 (black vertex), which contains four cited articles with three citation types.

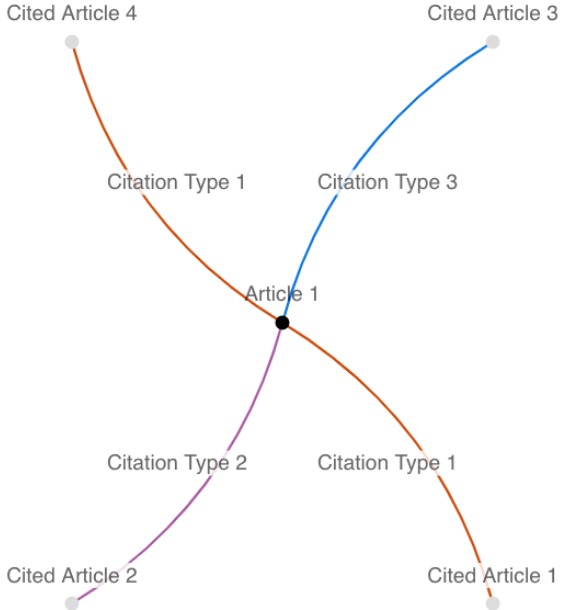

Figure 1: Citation network with citation types. Edges are colored based on the citation type, and cited articles appear as the neighborhood of the center vertex.

Let T0, T1, and T2 be shorthands for Table 1, Table 2, and Table 3, respectively. In the following, we propose a few prototypical user flows for how a researcher might answer Q1-Q5 by retrieving articles using the following citation types:

1. (Q1) **Evolutionary** (T1), **CoCoGM** (T2), or **Fundamental idea** (T3), since the current article is based on or compared against the cited article.

2. (Q2) **Background/technical basis** (T3), which implies that the cited article contains background high-level/technical information.

3. (Q3) **Organic** (T1) or **PModi** (T2), which implies that the current article builds upon the ideas in the cited article.

4. (Q4) **CoCoXY/PSim** (T2) or **Fundamental idea** (T3), which implies that the citing article extends the current article (note that this requires a lookup of which articles are citing the current article).

5. (Q5) **PSim** (T2) or **Comparison** (T3), which implies similarities between the current and the cited article.

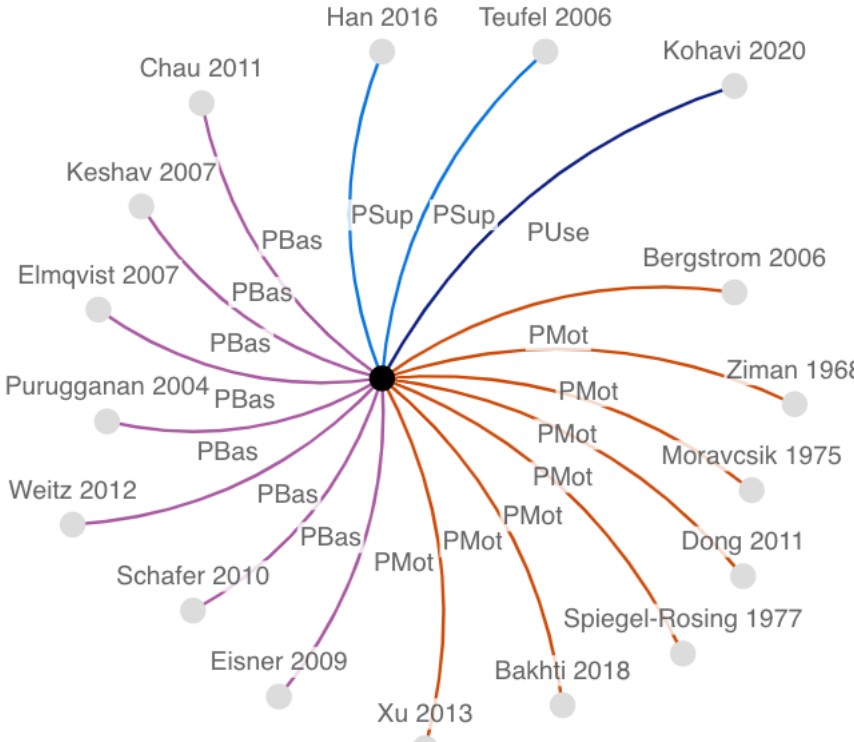

Figure 2: Citation network of this article. The citation types of Table 2 are used to manually classify every citation. Note that the D3 visualization of this network is more dynamic as hovering over the network elements reveals useful metadata and hyperlinks to articles that are embedded in the vertices.

While in the above user flows we decided which citation types to use heuristically, we argue that the above lookup process helps with answering Q1-Q5 in a timely manner. In order to demonstrate this idea, we build a rudimentary solution that uses this article as the center of the network. First, we construct this article's citation network neighborhood and derived the citation types based on T2 (Figure 2). Then we build and host a D3 visualization[6]. Lastly, in order to make the visualization immediately accessible from this article, we provide a shortened URL and a QR code (See Section 7).

### 4.2 AUGMENTING CITATION TYPES WITH AUTHOR COMMENTS

While we present citation types as an approach to answering Q1-Q5, one key limitation of the solution is that it lacks general expressiveness. In Figure 2, we observe relatively large numbers of citations of the same types (**PBas** and **PMot**). This makes it difficult for readers to differentiate between references of the same citation type.

To supplement the citation visualization, we further propose a workflow where authors are given the option to input a free-form comment for every reference. This enables authors to add information that is not conveyable by the generic citation types. We anticipate that the literature review sections of articles will especially benefit from such comments since there could be many citations of the same type. Therefore, this additional expressiveness can help address the deluge of citations problem mentioned in Section 3.

---

[6]See `https://d3js.org` (retrieved on March 1, 2021).

In the dynamic network visualization that we propose (see Section 7), comments are displayed when hovering over the edges. We have provided some example comments for the following references: Chau et al. (2011) and Ziman (1968).

## 5    QUESTIONS AND FUTURE EXPLORATIONS

A few natural questions arise stated here to motivate future explorations:

**Citation types**

1. **Which citation type schema should be used?** Using a standardized schema allows global statistics of the network to be computed (Schäfer & Kasterka, 2010), such as "Return past articles that are similar to Article 0". This can be obtained, for example, by using **PSim** (T2). Moreover, a standardized schema will minimize the effort needed by readers when navigating the citation networks. We conjecture that the most expressive schema would work best (e.g., T2), and envision that the schema will rarely evolve as long as the typical construct of a scientific article remains the same.

2. **Who should be performing the citation type classification?** While article authors know their article the best, different authors may interpret the citation types differently, even though they are citing a article similarly. Third-party annotators (Schäfer & Kasterka (2010) proposed crowd-based classification), who are more likely less familiar than the authors, might be able to more consistently classify the citations, as they might perform the task more often than individual authors. Lastly, automated techniques (Xu et al., 2013; Bakhti et al., 2018) can perform classification in a scalable manner. We think that moving forward, the most effective way to get started and evaluate these visualizations is if authors themselves could include these visualizations along with their articles. In our proposal, the visualizations are hosted separately, so they could evolve and are version-controlled. For past articles, automated techniques could be a great approach.

3. **What are good properties of citation types?** When defining citation types with the purpose of searches in the citation graph, properties such as transitivity would be helpful. For example, if **PSim** is transitive, then if A cites B with type **PSim**, and B cites C with type **PSim**, then if C were to be cited in A, it would also be cited with type **PSim**. Exploration of such properties for specific citation types could lead to useful global properties of the citation network.

**Citation network and visualization**

1. **What is the best way of providing the visualization?** Is a shortened URL presented as a footnote of the references section the best article element for accessing the visualization? Other options include embedding a QR code or the visualization itself in the article. However, embedding the visualization directly will remove the dynamism of the visualization.

2. **What is the easiest way for authors to include the visualization?** We envision a workflow that includes a plugin for the (raw) bibliography section of the article. Here, authors could assign citation types to their bibliographic entries, and work with a third party service that automatically creates and hosts a visualization.

3. **Should higher-order neighborhoods be included?** Our interviewees often mentioned that they wanted to quickly find out which article is cited and why it is cited. This dovetails with viewing the first-order neighborhood of the network. However, Q1, Q2, and Q5 could benefit from higher-order neighborhoods, as those questions involve more global information of the network. Understanding of use cases for such higher order neighborhoods are related to understanding the properties of citation types (Point 3 of the 'Citation types' section above). In the current form, we think that while there is a benefit from having second and third-order neighborhoods, the reader will encounter diminishing returns since the visualization would be less effective as it would contain many edges. Further exploration on visualization types will need to be conducted to enable higher-order visualizations.

4. **How can we evaluate the effectiveness of the visualizations?** If the visualizations are served via the reader's web browser, then we can apply web measurement techniques (Kohavi et al., 2020) to understand aggregated user behavior. Furthermore, we can compare different visualization aspects, such as citation schema, graph layout, colour palette, existence of legend (which may clutter the responsive mobile view), expressing richer semantics on the edges (e.g., bolder lines to indicate stronger dependence), etc.

**Augmenting citation types with comments**

1. **How can one measure the efficacy of providing comments?** While authors' comments can increase the expressiveness of information provided per reference, they also increase the amount of information being stored in the network visualization. In order to understand the impact of comments, we think that metrics like (1) authors' likelihood to input a comment, and (2) users' likelihood to engage with a reference in the network visualization given that it has a comment, can be used to measure efficacy.

2. **How can we increase authors' likelihood to input a comment?** If we can prove that authors' comments are helpful for readership (see the previous question), we believe that there are two design goals to achieve: (1) make the workflow for adding comments as simple as possible, and (2) develop incentives for authors to add comments. For (2), a possible incentive is a way for authors to refer back to their own comments on references while they are developing the literature review section.

## 6 ACCESSIBILITY STATEMENT

We commit to providing solutions that are accessible to the widest possible audience. We recognise the importance of making research accessible to people with a diverse range of hearing, movement, sight, and cognitive abilities.

The solution we propose is inherently aligned with improving the dissemination of information by diversifying the format available for reference sections that are typically exclusively presented in a flat textual format. Article readers are given an alternative way to interact with and visualize reference sections, raising the standard of the article reading user experience (UX) for researchers.

The accompanying visualization tool to this solution is web-based and enables users to:

- change colours and contrast levels;
- zoom in and out to the desired size;
- access the graphic through a variety of devices including mobile phones and tablets.

In addition, the colour palette used is Web Accessibility Content Guidelines (WCAG) 2.1 Level AA compliant. It was a deliberate choice to adhere to Level AA rather than Level AAA as the primary objective of colouring the edges of the citation network is to distinguish between colours. Level AAA would have not provided the necessary contrast between colours. Font, headings, and structure adhere to guidelines provided by the British Dyslexia Association based on WCAG 2.1.

**How accessible is the website?** The website solution is currently a prototype and by definition has minimal features. Future improvements could be made to accessibility depending on uptake of the solution. These may include the ability to:

- navigate most of the website using just a keyboard;
- navigate most of the website using speech recognition software;
- listen to most of the website using a screen reader (including the most recent versions of JAWS, NVDA, and VoiceOver);
- generate article copies of the visual.

These are currently limitations to the solution. New accessibility features will be considered and prioritized based on user feedback.

**Feedback and contact information**  To give feedback on accessibility and to get more information, please contact p.manggala@uva.nl.

ACKNOWLEDGMENTS

We thank all the researchers we interviewed. We are grateful to two anonymous referees for many helpful comments.

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

## 7 REFERENCES ADD-ON

Citation graph URL

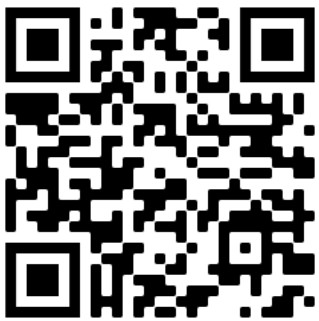

Figure 3: Citation graph URL behind a QR code.

