# OpenReview forum: "On augmenting the references section with a citation network visualization"
_ICLR.cc/2021/Workshop/Rethinking_ML_Papers — Rethinking ML Papers - ICLR 2021 workshop Poster_

### Official Review · AnonReviewer2 · 2021-03-29
**A good topic to discuss in a workshop format**

**Accessibility:**

Score of 4 (Strong): Submission states accessibility concerns and provides solutions within the proposed framework. However, it does not declare the limitations and exceptions.

**Litreview:**

Score of 3 (Neutral): The submission acknowledges previous work, but does not necessarily explain how the submission differentiates itself (i.e we want to avoid the “deluge of citation” strategy, leaving the reviewer to click through references and figure this part out for themselves).

**Problemstatement:**

Score of 4 (Strong): The submission sets a very strong example of how to address the problem, which should be relevant to the workshop themes.

**Relevance:**

Score of 5 (Exceptional): Like (4) but does so with multiple themes of the workshop.

**Results:**

Score of 3 (Neutral): Submission is well designed and provides a good level of coherency/novelty/interactivity.

**Reviewerconfidence:**

4 - I have spent some time thinking about similar issues and I think it's a good idea to discuss it in a ML context, but have not read all the cited/relevant literature about adding citation networks to papers.

**Reviewtext:**

The paper proposes to add citation networks to the references section of papers, and that this network integrates the type of citation (such as whether the current paper is extending another paper, general background, etc). The motivation is based on interviews conducted with practitioners in ML from different backgrounds and their experiences with reading literature.  The proposal is for authors to include the networks themselves.

This is a overall nice idea that fits the workshop. The implementation of how the network is presented can probably be improved (some suggestions below), but the workshop seems a perfect opportunity to discuss that.

Some ideas (in general, not about whether the submission is relevant for the workshop):

- The authors list several attempts to visualize networks in Section 1.1., but do not elaborate - it would be useful to know whether these ideas were adopted anywhere, or if not, why not, etc.

- The visualization of the citation types is not so intuitive to me, it would be great to include a legend in the plots and maybe vary the line type (bolder lines to indicate stronger dependence, for example)

- It might be interesting to discuss including citation types, or including a citation network, separately. Would one be easier to adopt than the other, or do they fail without each other?


**Score:**

Accept: The reviewer believes the submission provides a novel and reliable scheme to improve science communication but needs improvement.

---

### Official Review · AnonReviewer1 · 2021-03-29
**Visualizing citation provenance and type in research papers**

**Accessibility:**

Score of 4 (Strong): Submission states accessibility concerns and provides solutions within the proposed framework. However, it does not declare the limitations and exceptions.

**Litreview:**

Score of 4 (Strong): The submission directly differentiates itself from previous works and formats.

**Problemstatement:**

Score of 4 (Strong): The submission sets a very strong example of how to address the problem, which should be relevant to the workshop themes.

**Relevance:**

Score of 5 (Exceptional): Like (4) but does so with multiple themes of the workshop.

**Results:**

Score of 4 (Strong): Submission is very well structured and follows all the criteria (i.e. clarity, novelty, interactivity, and coherency). However, practical significance/theoretical implications are not discussed.

**Reviewerconfidence:**

My confidence in my review is in between a 4 and 5. I have familiarity in improving research paper readability both from a practical research perspective and from a tool builder's perspective. I really like the larger ideas here, but think the prototype and implementation could be polished.

**Reviewtext:**

The proposed work on augmenting the references section with a citation network visualization aims to improve research paper readability through a network graph visualization that captures information about a paper’s reference types. References are a standardized practice in scientific publishing across fields, and despite their ubiquity and usefulness, remain underexplored. Categorizing references as different types within a paper, and visualizing a paper’s citation network alongside the references is an interesting solution to leverage a paper’s reference provenance, present how author’s view different citations, and help readers identify relevant citations faster. The proposed prototype, while a first iteration, is sufficient for convincing another researcher of the importance and usefulness of larger ideas here. In short, it is enough to “get it.” I would enjoy seeing a complete design process and implementation around this idea. The accessibility discussion is also welcomed and its ties together well given the main idea is to make research easier to navigate.

My only suggestion to improve the work is to think about the structure of the citation graphs. Of the two graphs presented in the work, both are “star” graphs (1 internal node, many leaf nodes). There could be more interesting and complex structure in the citation graph if it also included edges between other nodes that were not the original paper. Otherwise, every citation graph would be a star, which is likely not the best visualization to choose for this task, i.e., a colored and sortable table may be easier to use.

One previous work on a mixed-initiative system guides readers interactively exploring citation graphs would be worth including in the related work.

Duen Horng Chau, Aniket Kittur, Jason I. Hong, and Christos Faloutsos. “Apolo: making sense of large network data by combining rich user interaction and machine learning.” ACM CHI. 2011.

**Score:**

Accept: The reviewer believes the submission provides a novel and reliable scheme to improve science communication but needs improvement.

---

### Meta-Review · Area_Chair1 · 2021-04-01

**Recommendation:** Accept
**Confidence:** 4

**Metareview:**

This work proposes adding a graph of cited papers in the reference section, which I found a great contribution to the accessibility and inclusivity in our research community. This will very much help the early career researchers to explore the field and speed up the literature review.

As mentioned by one of the reviewers, there can be further improvements on the type and the structure of citation graphs, and how that can be implemented effectively. In the current implementation every citation graph could appear in the same visualization format (most likely a star, as also pointed out by reviewer 1) which might not be the best arrangement for the studied subject.

In addition, the figures in the paper can be better visualized, a legend describing edges can help readers to better understand the citation network. Another concern is that the authors did not discuss the weaknesses and limitations to the proposed approach and that can be refined. The accessibility statement is yet well declared and clarified.

Overall, I believe that this work provides a novel scheme to better develop scientific articles. Hence, I recommend this work to be presented at the workshop.

---

### Decision · Program_Chairs · 2021-04-01

Accept (Poster)